# Algorithms and matching lower bounds for approximately-convex optimization

**Yuanzhi Li**
Department of Computer Science
Princeton University
Princeton, NJ, 08450
yuanzhil@cs.princeton.edu

**Andrej Risteski**
Department of Computer Science
Princeton University
Princeton, NJ, 08450
risteski@cs.princeton.edu

## Abstract

In recent years, a rapidly increasing number of applications in practice requires optimizing non-convex objectives, like training neural networks, learning graphical models, maximum likelihood estimation. Though simple heuristics such as gradient descent with very few modifications tend to work well, theoretical understanding is very weak.

We consider possibly the most natural class of non-convex functions where one could hope to obtain provable guarantees: functions that are "approximately convex", i.e. functions $\tilde{f} : \mathbb{R}^d \to \mathbb{R}$ for which there exists a *convex function* $f$ such that for all $x$, $|\tilde{f}(x) - f(x)| \le \Delta$ for a fixed value $\Delta$. We then want to minimize $\tilde{f}$, i.e. output a point $\tilde{x}$ such that $\tilde{f}(\tilde{x}) \le \min_x \tilde{f}(x) + \epsilon$.

It is quite natural to conjecture that for fixed $\epsilon$, the problem gets harder for larger $\Delta$, however, the exact dependency of $\epsilon$ and $\Delta$ is not known. In this paper, we significantly improve the known lower bound on $\Delta$ as a function of $\epsilon$ and an algorithm matching this lower bound for a natural class of convex bodies. More precisely, we identify a function $T : \mathbb{R}^+ \to \mathbb{R}^+$ such that when $\Delta = O(T(\epsilon))$, we can give an algorithm that outputs a point $\tilde{x}$ such that $\tilde{f}(\tilde{x}) \le \min_x \tilde{f}(x) + \epsilon$ within time $poly\left(d, \frac{1}{\epsilon}\right)$. On the other hand, when $\Delta = \Omega(T(\epsilon))$, we also prove an *information theoretic* lower bound that any algorithm that outputs such a $\tilde{x}$ must use *super polynomial* number of evaluations of $\tilde{f}$.

## 1 Introduction

Optimization of convex functions over a convex domain is a well studied problem in machine learning, where a variety of algorithms exist to solve the problem efficiently. However, in recent years, practitioners face ever more often non-convex objectives – e.g. training neural networks, learning graphical models, clustering data, maximum likelihood estimation etc. Albeit simple heuristics such as gradient descent with few modifications usually work very well, theoretical understanding in these settings are still largely open.

The most natural class of non-convex functions where one could hope to obtain provable guarantees is functions that are "approximately convex": functions $\tilde{f} : \mathbb{R}^d \to \mathbb{R}$ for which there exists a *convex function* $f$ such that for all $x$, $|\tilde{f}(x) - f(x)| \le \Delta$ for a fixed value $\Delta$. In this paper, we focus on *zero order optimization* of $\tilde{f}$: an algorithm that outputs a point $\tilde{x}$ such that $\tilde{f}(\tilde{x}) \le \min_x \tilde{f}(x) + \epsilon$, where the algorithm in the course of its execution is allowed to pick points $x \in \mathbb{R}^d$ and query the value of $\tilde{f}(x)$.

Trivially, one can solve the problem by constructing a $\epsilon$-net and search through all the net points. However, such an algorithm requires $\Omega\left(\frac{1}{\epsilon}\right)^d$ evaluations of $\tilde{f}$, which is highly inefficient in high dimension. In this paper, we are interested in *efficient algorithms*: algorithms that run in time $poly\left(d, \frac{1}{\epsilon}\right)$ (in particular, this implies the algorithm makes $poly\left(d, \frac{1}{\epsilon}\right)$ evaluations of $\tilde{f}$).

One extreme case of the problem is $\Delta = 0$, which is just standard convex optimization, where algorithms exist to solve it in polynomial time for every $\epsilon > 0$. However, even when $\Delta$ is any quantity $> 0$, none of these algorithms extend without modification. (Indeed, we are not imposing *any* structure on $\tilde{f} - f$ like stochasticity.) Of course, when $\Delta = +\infty$, the problem includes any non-convex optimization, where we cannot hope for an efficient solution for any finite $\epsilon$. Therefore, the crucial quantity to study is the optimal tradeoff of $\epsilon$ and $\Delta$: For which $\epsilon, \Delta$ the problem can be solved in polynomial time, and for which it can not.

In this paper, we study the rate of $\Delta$ as a function of $\epsilon$: We identify a function $T : \mathbb{R}^+ \to \mathbb{R}^+$ such that when $\Delta = O(T(\epsilon))$, we can give an algorithm that outputs a point $\tilde{x}$ such that $\tilde{f}(\tilde{x}) \leq \min_x \tilde{f}(x) + \epsilon$ within time $poly\left(d, \frac{1}{\epsilon}\right)$ over a natural class of *well-conditioned* convex bodies. On the other hand, when $\Delta = \tilde{\Omega}(T(\epsilon))$[1], we also prove an *information theoretic* lower bound that any algorithm outputs such $\tilde{x}$ must use *super polynomial* number of evaluations of $\tilde{f}$. Our result can be summarized as the following two theorems:

**Theorem** (Algorithmic upper bound, informal). *There exists an algorithm $\mathcal{A}$ that for any function $\tilde{f}$ over a* well-conditioned *convex set in $\mathbb{R}^d$ of diameter 1 which is $\Delta$ close to an* 1-Lipschitz convex function [2] *f, and*

$$\Delta = O\left(\max\left(\left\{\frac{\epsilon^2}{\sqrt{d}}, \frac{\epsilon}{d}\right\}\right)\right)$$

*$\mathcal{A}$ finds a point $\tilde{x}$ such that $\tilde{f}(\tilde{x}) \leq \min_x \tilde{f}(x) + \epsilon$ within time $poly\left(d, \frac{1}{\epsilon}\right)$*

The notion of well-conditioning will formally be defined in section 3, but intuitively captures the notion that the convex body "curves" in all directions to a good extent.

**Theorem** (Information theoretic lower bound, informal). *For every algorithm $\mathcal{A}$, every $d, \Delta, \epsilon$ with*

$$\Delta = \tilde{\Omega}\left(\max\left\{\frac{\epsilon^2}{\sqrt{d}}, \frac{\epsilon}{d}\right\}\right)$$

*there exists a function $\tilde{f}$ on a convex set in $\mathbb{R}^d$ of diameter 1, and $\tilde{f}$ is $\Delta$ close to an* 1-Lipschitz convex function *f, such that $\mathcal{A}$ can not find a point $\tilde{x}$ with $\tilde{f}(\tilde{x}) \leq \min_x \tilde{f}(x) + \epsilon$ in $poly\left(d, \frac{1}{\epsilon}\right)$ evaluations of $\tilde{f}$.*

## 2 Prior work

To the best of our knowledge, there are three works on the problem of approximately convex optimization, which we summarize briefly below.

On the algorithmic side, the classical paper by [DKS14] considered optimizing *smooth* convex functions over convex bodies with *smooth* boundaries. More precisely, they assume a bound on both the gradient and the Hessian of $F$. Furthermore, they assume that for every small ball centered at a point in the body, a large proportion of the volume of the ball lies in the body. Their algorithm is local search: they show that for a sufficiently small $r$, in a ball of radius $r$ there is with high probability a point which has a smaller value than the current one, as long as the current value is sufficiently larger than the optimum. For *constant-smooth* functions only, their algorithm applies when $\Delta = O(\frac{\epsilon}{\sqrt{d}})$.

Also on the algorithmic side, the work by [BLNR15] considers 1-Lipschitz functions, but their algorithm only applies to the case where $\Delta = O(\frac{\epsilon}{d})$ (so not optimal unless $\epsilon = O(\frac{1}{\sqrt{d}})$). Their methods rely on sampling log-concave distribution via hit and run walks. The crucial idea is to show that for approximately convex functions, one needs to sample from "approximately log-concave"

distributions, which they show can be done by a form of rejection sampling together with classical methods for sampling log-concave distributions.

Finally, [SV15] consider information theoretic lower bounds. They show that when $\Delta = 1/d^{1/2-\delta}$ no algorithm can, in polynomial time, achieve achieve $\epsilon = \frac{1}{2} - \delta$, when optimizing a convex function over the hypercube. This translates to a super polynomial information theoretic lower bound when $\Delta = \Omega(\frac{\epsilon}{\sqrt{d}})$. They additionally give lower bounds when the approximately convex function is multiplicatively, rather than additively, close to a convex function. [3]

We also note a related problem is zero-order optimization, where the goal is to minimize a function we only have value oracle access to. The algorithmic motivations here come from various applications where we only have black-box access to the function we are optimizing, and there is a classical line of work on characterizing the oracle complexity of convex optimization.[NY83, NS, DJWW15]. In all of these settings however, the oracles are either noiseless, or the noise is stochastic, usually because the target application is in bandit optimization. [AD10, AFH$^+$11, Sha12]

# 3   Overview of results

Formally, we will consider the following scenario.

**Definition 3.1.** *A function* $\tilde{f} : \mathcal{K} \to \mathbb{R}^d$ *will be called* $\Delta$-approximately convex *if there exists a 1-Lipschitz convex function* $f : \mathcal{K} \to \mathbb{R}^d$, *s.t.* $\forall x \in \mathcal{K}, |\tilde{f}(x) - f(x)| \leq \Delta$.

For ease of exposition, we also assume that $\mathcal{K}$ has diameter $1$[4]. We consider the problem of *optimizing* $\tilde{f}$, more precisely, we are interesting in finding a point $\tilde{x} \in \mathcal{K}$, such that
$$\tilde{f}(\tilde{x}) \leq \min_{x \in \mathcal{K}} \tilde{f}(x) + \epsilon$$

We give the following results:

**Theorem 3.1** (Information theoretic lower bound)**.** *For very constant* $c \geq 1$, *there exists a constant* $d_c$ *such that for every algorithm* $\mathcal{A}$, *every* $d \geq d_c$, *there exists a convex set* $\mathcal{K} \subseteq \mathbb{R}^d$ *with diameter 1, an* $\Delta$-*approximate convex function* $\tilde{f} : \mathcal{K} \to \mathbb{R}$ *and* $\epsilon \in [0, 1/64)$ [5] *such that*
$$\Delta \geq \max\left\{\frac{\epsilon^2}{\sqrt{d}}, \frac{\epsilon}{d}\right\} \times \left(13c \log \frac{d}{\epsilon}\right)^2$$

*Such that* $\mathcal{A}$ *fails to output, with probability* $\geq 1/2$, *a point* $\tilde{x} \in \mathcal{K}$ *with* $\tilde{f}(\tilde{x}) \leq \min_{x \in \mathcal{K}}\{\tilde{f}(x)\} + \epsilon$ *in* $o((\frac{d}{\epsilon})^c)$ *time.*

In order to state the upper bounds, we will need the definition of a well-conditioned body:

**Definition 3.2** ($\mu$-well-conditioned)**.** *A convex body* $\mathcal{K}$ *is said to be* $\mu$-well-conditioned *for* $\mu \geq 1$, *if there exists a function* $F : \mathbb{R}^d \to \mathbb{R}$ *such that* $\mathcal{K} = \{x | F(x) \leq 0\}$ *and for every* $x \in \partial\mathcal{K}$: $\frac{\|\nabla^2 F(x)\|_2}{\|\nabla F(x)\|_2} \leq \mu$.

This notion of well-conditioning of a convex body to the best of our knowledge has not been defined before, but it intuitively captures the notion that the convex body should "curve" in all directions to a certain extent. In particular, the unit ball has $\mu = 1$.

**Theorem 3.2** (Algorithmic upper bound)**.** *Let* $d$ *be a positive integer,* $\delta > 0$ *be a positive real number,* $\epsilon, \Delta$ *be two positive real number such that*
$$\Delta \leq \max\left\{\frac{\epsilon^2}{\mu\sqrt{d}}, \frac{\epsilon}{d}\right\} \times \frac{1}{16348}$$

*Then there exists an algorithm* $\mathcal{A}$ *such that on given any* $\Delta$-*approximate convex function* $\tilde{f}$ *over a* $\mu$-*rounded convex set* $\mathcal{K} \subseteq \mathbb{R}^d$ *of diameter 1,* $\mathcal{A}$ *returns a point* $\tilde{x} \in \mathcal{K}$ *with probability* $1 - \delta$ *in time* $poly\left(d, \frac{1}{\epsilon}, \log\frac{1}{\delta}\right)$ *such that*
$$\tilde{f}(\tilde{x}) \leq \min_{x \in \mathcal{K}} \tilde{f}(x) + \epsilon$$

For the reader wishing to digest a condition-free version of the above result, the following weaker result is also true (and much easier to prove):

**Theorem 3.3** (Algorithmic upper bound (condition-free)). *Let $d$ be a positive integer, $\delta > 0$ be a positive real number, $\epsilon, \Delta$ be two positive real number such that*

$$\Delta \leq \max\left\{\frac{\epsilon^2}{\sqrt{d}}, \frac{\epsilon}{d}\right\} \times \frac{1}{16348}$$

*Then there exists an algorithm $\mathcal{A}$ such that on given any $\Delta$-approximate convex function $\tilde{f}$ over a $\mu$-rounded convex set $\mathcal{K} \subseteq \mathbb{R}^d$ of diameter 1, $\mathcal{A}$ returns a point $\tilde{x} \in \mathcal{K}$ with probability $1 - \delta$ in time poly $\left(d, \frac{1}{\epsilon}, \log\frac{1}{\delta}\right)$ such that*

$$\tilde{f}(\tilde{x}) \leq \min_{x \in S(\mathcal{K}, -\epsilon)} \tilde{f}(x) + \epsilon$$

*Where $S(\mathcal{K}, -\epsilon) = \{x \in \mathcal{K} | \mathbb{B}_\epsilon(x) \subseteq \mathcal{K}\}$*

The result merely states that we can output a value that competes with points "well-inside" the convex body – around which a ball of radius of $\epsilon$ still lies inside the body.

The assumptions on the diameter of $\mathcal{K}$ and the Lipschitz condition are for convenience of stating the results. It's quite easy to extend both the lower and upper bounds to an arbitrary diameter and Lipschitz constant, as we discuss in Section 6.

### 3.1 Proof techniques

We briefly outline the proof techniques we use. We proceed with the information theoretic lower bound first. The idea behind the proof is the following. We will construct a function $G(x)$ and a family of convex functions $\{f_w(x)\}$ depending on a direction $w \in \mathcal{S}^d$ ($\mathcal{S}^d$ is the unit sphere in $\mathbb{R}^d$). On one hand, the minimal value of $G$ and $f_w$ are quite different: $\min_x G(x) \geq 0$, and $\min_x f_w(x) \leq -2\epsilon$. On the other hand, the approximately convex function $\tilde{f}_w(x)$ for $f_w(x)$ we consider will be such that $\tilde{f}_w(x) = G(x)$ except in a very small cone around $w$. Picking $w$ at random, no algorithm with small number of queries will, with high probability, every query a point in this cone. Therefore, the algorithm will proceed as if the function is $G(x)$ and fail to optimize $\tilde{f}_w$.

Proceeding to the algorithmic result, since [BLNR15] already shows the existence of an efficient algorithm when $\Delta = O(\frac{\epsilon}{d})$, we only need to give an algorithm that solves the problem when $\Delta = \Omega(\frac{\epsilon}{d})$ and $\Delta = O(\frac{\epsilon^2}{\sqrt{d}})$ (i.e. when $\epsilon, \Delta$ are large). There are two main ideas for the algorithm. First, we show that the gradient of a *smoothed* version of $\tilde{f}_w$ (in the spirit of [FKM05]) at any point $x$ will be correlated with $x^* - x$, where $x^* = \operatorname{argmin}_{x \in \mathcal{K}} \tilde{f}_w(x)$. The above strategy will however require averaging the value of $\tilde{f}_w$ along a ball of radius $\epsilon$, which in many cases will not be contained in $\mathcal{K}$ (especially when $\epsilon$ is large). Therefore, we come up with a way to *extend* $\tilde{f}_w$ outside of $\mathcal{K}$ in a manner that maintains the correlation with $x^* - x$.

## 4 Information-theoretic lower bound

In this section, we present the proof of Theorem 3.1.

The idea is to construct a function $G(x)$, a family of convex functions $\{f_w(x)\}$ depending on a direction $w \in \mathcal{S}^d$, such that $\min_x G(x) \geq 0$, $\min_x f_w(x) \leq -2\epsilon$, and an approximately convex $\tilde{f}_w(x)$ for $f_w(x)$ such that $\tilde{f}_w(x) = G(x)$ except in a very small "critical" region depending on $w$. Picking $w$ at random, we want to argue that the algorithm will with high probability not query the critical region. The convex body $\mathcal{K}$ used in the lower bound will be arguably the simplest convex body imaginable: the unit ball $\mathbb{B}_1(0)$.

We might hope to prove a lower bound for even a linear function $f_w$ for a start, similarly as in [SV15]. A reasonable candidate construction is the following: we set $f_w(x) = -\epsilon\langle w, x\rangle$ for some random chosen unit vector $w$ and define $\tilde{f}(x) = 0$ when $|\langle x, w\rangle| \leq \frac{\log\frac{d}{\epsilon}}{\sqrt{d}}\|x\|_2$ and $\tilde{f}(x) = f_w(x)$ otherwise.[6]

Observe, this translates to $\Delta = \frac{\log \frac{d}{\epsilon}}{\sqrt{d}}\epsilon$. It's a standard concentration of measure fact that for "most" of the points $x$ in the unit ball, $|\langle x, w \rangle| \leq \frac{\log \frac{d}{\epsilon}}{\sqrt{d}}\|x\|_2$. This implies that any algorithm that makes a polynomial number of queries to $\tilde{f}$ will with high probability see 0 in all of the queries, but clearly $\min \tilde{f}(x) = -\epsilon$. However, this idea fails to generalize to optimal range as $\Delta = \frac{1}{\sqrt{d}}\epsilon$ is tight for linear, even smooth functions.[7]

In order to obtain the optimal bound, we need to modify the construction to a non-linear, non-smooth function. We will, in a certain sense, "hide" a random linear function inside a non-linear function. For a random unit vector $w$, we consider two regions inside the unit ball: a core $\mathcal{C} = \mathbb{B}_r(0)$ for $r = \max\{\epsilon, \frac{1}{\sqrt{d}}\}$, and a "critical angle" $\mathcal{A} = \{x \mid |\langle x, w\rangle| \geq \frac{\log \frac{d}{\epsilon}}{\sqrt{d}}\|x\|_2\}$. The *convex* function $f$ will look like $\|x\|_2^{1+\alpha}$ for some $\alpha > 0$ outside $\mathcal{C} \cup \mathcal{A}$ and $-\epsilon\langle w, x \rangle$ for $x \in \mathcal{C} \cup \mathcal{A}$. We construct $\tilde{f}$ as $\tilde{f} = f$ when $f(x)$ is sufficiently large (e.g. $|f(x)| > \frac{\Delta}{2}$) and $\frac{\Delta}{2}$ otherwise. Clearly, such $\tilde{f}$ obtain its minimal at point $w$, with $\tilde{f}(w) = -\epsilon$. However, since $\tilde{f} = \|x\|_2^{1+\alpha}$ outside $\mathcal{C}$ or $\mathcal{A}$, the algorithm needs either query $\mathcal{A}$ or query $\mathcal{C} \cap \mathcal{A}^c$ to detect $w$. The former happens with exponentially small probability in high dimensions, and for any $x \in \mathcal{C} \cap \mathcal{A}^c$, $|f(x)| = \epsilon|\langle w, x \rangle| \leq \frac{\epsilon \log \frac{d}{\epsilon}}{\sqrt{d}}\|x\|_2 \leq \frac{\epsilon \log \frac{d}{\epsilon}}{\sqrt{d}}r \leq \max\{\frac{\epsilon^2}{\sqrt{d}}, \frac{\epsilon}{d}\} \times \log \frac{d}{\epsilon} \leq \frac{\Delta}{2}$, which implies that $\tilde{f}(x) = \frac{\Delta}{2}$. Therefore, the algorithm will fail with high probability.

Now, we move on to the detailed of the constructions. We will consider $\mathcal{K} = \mathbb{B}_{\frac{1}{2}}(0)$: the ball of radius $\frac{1}{2}$ in $\mathbb{R}^d$ centered at 0. [8]

## 4.1 The family $\{f_w(x)\}$

Before delving into the construction we need the following definition:

**Definition 4.1** (Lower Convex Envelope (LCE)). *Given a set $\mathcal{S} \subseteq \mathbb{R}^d$, a function $F : \mathcal{S} \to \mathbb{R}$, define the lower convex envelope $F_{LCE} = LCE(F)$ as a function $F_{LCE} : \mathbb{R}^d \to \mathbb{R}$ such that for every $x \in \mathbb{R}^d$,*

$$F_{LCE}(x) = \max_{y \in \mathcal{S}}\{\langle x - y, \nabla F(y)\rangle + F(y)\}$$

**Proposition 4.1.** *$LCE(F)$ is convex.*

*Proof.* LCE(F) is the pointwise maximum of linear functions, so the claim follows. $\square$

**Remark** : The LCE of a function $F$ is a function defined over the entire $\mathbb{R}^d$, while the input function $F$ is only defined in a set $\mathcal{S}$ (not necessarily convex set). When the input function $F$ is convex, LCE$(F)$ can be considered as an extension of $F$ to the entire $\mathbb{R}^d$.

To define the family $f_w(x)$, we will need four parameters: a power factor $\alpha > 0$, a shrinking factor $\beta$, and a radius factor $\gamma > 0$, and a vector $w \in \mathbb{R}^d$ such that $\|w\|_2 = \frac{1}{2}$, which we specify in a short bit.

**Construction 4.1.** *Given $w, \alpha, \beta, \gamma$, define the core $\mathcal{C} = \mathbb{B}_\gamma(0)$, the critical angle $\mathcal{A} = \{x \mid |\langle x, w \rangle| \geq \beta\|x\|_2\}$ and let $\mathcal{H} = \mathcal{K} \cap \overline{\mathcal{C}} \cap \overline{\mathcal{A}}$. Let $\tilde{h} : \mathcal{H} \to \mathbb{R}$ be defined as*

$$\tilde{h}(x) = \frac{1}{2}\|x\|_2^{1+\alpha}$$

*and define $l_w(x) = -8\epsilon\langle x, w \rangle$. Finally let $f_w : \mathcal{K} \to \mathbb{R}^d$ as*

$$f_w(x) = \max\left\{\tilde{h}_{LCE}(x), l_w(x)\right\}$$

*Where $\tilde{h}_{LCE} = LCE(\tilde{h})$ as in Definition 4.1.*

We then construct the "hard" function $\tilde{f}_w$ as the following:

**Construction 4.2.** *Consider the function $\tilde{f}_w : \mathcal{K} \to \mathbb{R}$:*

$$\tilde{f}_w(x) = \begin{cases} f_w(x) & \text{if } x \in \mathcal{K} \cap (\overline{\mathcal{C}} \cup \mathcal{A}) \text{ ;} \\ \max\{f_w(x), \frac{1}{2}\Delta\} & \text{otherwise.} \end{cases}$$

Consider the following settings of the parameters $\beta, \gamma, \alpha$ (depending on the magnitude of $\epsilon$):

- Case 1, $\frac{1}{\sqrt{d}} \leq \epsilon \leq \frac{1}{(\log d)^2}$: $\beta = \frac{\sqrt{c \log \frac{d}{\epsilon}}}{\sqrt{d}}$, $\gamma = 10c\epsilon(\log \frac{d}{\epsilon})^{1.5}$, $\alpha = \frac{1}{\log(1/\gamma)}$.

- Case 2, $\epsilon \leq \frac{1}{\sqrt{d}}$: $\beta = \frac{\sqrt{c \log d/\epsilon}}{\sqrt{d}}$, $\gamma = \frac{10c}{\sqrt{d}}(\log d/\epsilon)^{3/2}$, $\alpha = \frac{1}{\log(1/\gamma)}$.

- Case 3, $\frac{1}{64} \geq \epsilon \geq \frac{1}{(\log d)^2}$: $\beta = \frac{\sqrt{c \log d}}{\sqrt{d}}$, $\gamma = \frac{1}{2}$, $\alpha = 1$.

Then, the we formalize the proof intuition from the previous section with the following claims.

Following the the proof outline, we first show the minimum of $f_w$ is small, in particular we will show $f_w(w) \leq -2\epsilon$.

**Lemma 4.1.** $f_w(w) = -2\epsilon$

Finally, we show that $\tilde{f}_w$ is indeed a $\Delta$-approximately convex, by showing $\forall x \in \mathcal{K}, |f_w - \tilde{f}_w| \leq \Delta$ and $f_w$ is 1-Lipschitz and convex.

**Proposition 4.2.** $\tilde{f}_w$ is a $\Delta$-approximately convex.

Next, we construct $G(x)$, which does not depend on $w$, we want to show that for an algorithm with small number of queries of $\tilde{f}_w$, it can not distinguish $f_w$ from this function.

**Construction 4.3.** Let $G : \mathcal{K} \to \mathbb{R}$ be defined as:
$$G(x) = \begin{cases} \max\left\{\frac{1+\alpha}{4}\|x\|_2 - \frac{\alpha}{4}\gamma, \frac{1}{2}\Delta\right\} & \text{if } x \in \mathcal{K} \cap \mathcal{C}; \\ \frac{1}{2}\|x\|_2^{1+\alpha} & \text{otherwise.} \end{cases}$$

The following is true:

**Lemma 4.2.** $G(x) \geq 0$ and $\{x \in \mathcal{K} \mid G(x) \neq \tilde{f}_w(x)\} \subseteq \mathcal{A}$

We show how Theorem 3.1 is implied given these statements:

*Proof of Theorem 3.1.* With everything prior to this set up, the final claim is somewhat standard. We want to show that no algorithm can, with probability $\geq \frac{1}{2}$, output a point $x$, s.t. $\tilde{f}_w(x) \leq \min_x \tilde{f}_w(x) + \epsilon$. Since we know that $\tilde{f}_w(x)$ agrees with $G(x)$ everywhere except in $\mathcal{K} \cap \mathcal{A}$, and $G(x)$ satisfies $\min_x G(x) \geq \min_x \tilde{f}_w(x) + \epsilon$, we only need to show that with high probability, any polynomial time algorithm will not query any point in $\mathcal{K} \cap \mathcal{A}$.

Consider a (potentially) randomized algorithm $A$, making random choices $R_1, R_2, \ldots, R_m$. Conditioned on a particular choice of randomness $r_1, r_2, \ldots, r_m$, for a random choice of $w$, each $r_i$ lies in $\mathcal{A}$ with probability at most $\exp(-c \log(d/\epsilon))$, by a standard Gaussian tail bound. Union bounding, since $m = o((\frac{d}{\epsilon})^c)$ for an algorithm that runs in time $o((\frac{d}{\epsilon})^c)$, the probability that at least of the queries of $A$ lies in $\mathcal{A}$ is at most $\frac{1}{2}$.

But the claim is true for any choice $r_1, r_2, \ldots, r_m$ of the randomness, by averaging, the claim holds for $r_1, r_2, \ldots, r_m$ being sampled according to the randomness of the algorithm.

$\square$

The proofs of all of the lemmas above have been ommited due to space constraints, and are included in the appendix in full.

## 5 Algorithmic upper bound

As mentioned before, the algorithm in [BLNR15] covers the case when $\Delta = O(\frac{\epsilon}{d})$, so we only need to give an algorithm when $\Delta = \Omega(\frac{\epsilon}{d})$ and $\Delta = O(\frac{\epsilon^2}{d})$. Our approach will not be making use of simulated annealing, but a more robust version of gradient descent. The intuition comes from [FKM05] who use estimates of the gradient of a convex function derived from Stokes' formula:

$$\mathbb{E}_{w \sim \mathcal{S}^d}\left[\frac{d}{r}f(x+rw)w\right] = \int_{\mathbb{B}} \nabla f(x) dx$$

where $w \sim \mathcal{S}^d$ denotes $w$ being a uniform sample from the sphere $\mathcal{S}^d$. Our observation is the gradient estimation is robust to noise if we instead use $\tilde{f}$ in the left hand side. Crucially, robust is *not* in the sense that it approximates the gradient of $f$, but it preserves the crucial property of the gradient of $f$ we need: $\langle -\nabla f(x), x^* - x \rangle \geq f(x) - f(x^*)$. In words, this means if we move $x$ at direction $-\nabla f(x)$ for a small step, then $x$ will be closer to $x^*$, and we will show the property is preserved by $\tilde{f}$ when $\Delta \leq \frac{\epsilon^2}{\sqrt{d}}$. Indeed, we have that:

$$\left\langle -\mathbb{E}_{w\sim\mathcal{S}^d}\left[\frac{d}{r}\tilde{f}(x+rw)w\right], x^* - x \right\rangle$$

$$\geq -\mathbb{E}_{w\sim\mathcal{S}^d}\left[\left\langle \frac{d}{r}f(x+rw)w, x^* - x \right\rangle\right] - \frac{d\Delta}{r}\mathbb{E}_{w\sim\mathcal{S}^d}\left[|\langle w, x^* - x \rangle|\right]$$

The usual [FKM05] calculation shows that

$$\mathbb{E}_{w\sim\mathcal{S}^d)}\left[\left\langle \frac{d}{r}f(x+rw)w, x^* - x \right\rangle\right] = \Omega\left(f(x) - f(x^*) - 2r\right)$$

and $\frac{d}{r}\Delta\mathbb{E}_{w\sim U(S^d)}\left[|\langle w, x^* - x \rangle|\right]$ is bounded by $O(\frac{\Delta\sqrt{d}}{r})$, since $\mathbb{E}_{w\sim U(S^d)}\left[|\langle w, x^* - x \rangle|\right] = O(\frac{1}{\sqrt{d}})$.

Therefore, we want $f(x) - f(x^*) - 2r \geq \dfrac{\Delta\sqrt{d}}{r}$ whenever $f(x) - f(x^*) \geq \epsilon$. Choosing the optimal parameter leads to $r = \frac{\epsilon}{4}$ and $\Delta \leq \frac{\epsilon^2}{\sqrt{d}}$.

This intuitive calculation basically proves the simple upper bound guarantee (Theorem 3.3). On the other hand, the argument requires sampling from a ball of radius $\Omega(\epsilon)$ around point $x$. This is problematic when $\epsilon > \frac{1}{\sqrt{d}}$: many convex bodies (e.g. the simplex, $L^1$ ball after rescaling to diameter one) will not contain a ball of radius even $\frac{1}{\sqrt{d}}$. The idea is then to make the sampling possible by "extending" $\tilde{f}$ outside of $\mathcal{K}$. Namely, we define a new function $g : \mathbb{R}^d \to \mathbb{R}$ such that ($\Pi_{\mathcal{K}}(x)$ is the projection of $x$ to $\mathcal{K}$)

$$g(x) = \tilde{f}(\Pi_{\mathcal{K}}(x)) + d(x, \mathcal{K})$$

$g(x)$ will not be in general convex, but we instead directly bound $\langle \mathbb{E}_{w\sim}\left[\frac{1}{r}g(x+rw)w\right], x - x^* \rangle$ for $x \in \mathcal{K}$ and show that it behaves like $\langle -\nabla f(x), x^* - x \rangle \geq f(x) - f(x^*)$.

---

**Algorithm 1** Noisy Convex Optimization

---

1: Input: A convex set $\mathcal{K} \subset \mathbb{R}^d$ with $\mathsf{diam}(\mathcal{K}) = 1$ and $0 \in \mathcal{K}$. A $\Delta$-approximate convex function $\tilde{f}$

2: Define: $g : \mathbb{R} \to \mathbb{R}$ as:
$$\tilde{g}(x) = \tilde{f}(\Pi_{\mathcal{K}}(x)) + d(x, \mathcal{K})$$
where $\Pi_{\mathcal{K}}$ is the projection to $\mathcal{K}$ and $d(x, \mathcal{K})$ is the Euclidean distance from $x$ to $\mathcal{K}$.

3: Initial: $x_1 = 0, r = \frac{\epsilon}{128\mu}, \eta = \frac{\epsilon^3}{4194304d^2}, T = \frac{8388608d^2}{\epsilon^4}$.

4: **for** $t = 1, 2, ...., T$ **do**

5:     Let $v_t = \tilde{f}(x_t)$.

6:     Estimate up to accuracy $\frac{\epsilon}{4194304}$ in $l_2$ norm (by uniformly randomly sample $w$):

$$g_t = \mathbb{E}_{w\sim\mathcal{S}^d}\left[\frac{d}{r}\tilde{g}(x_t + rw)w\right]$$

    where $w \sim \mathcal{S}^d$ means $w$ is uniform sample from the unit sphere.

7:     Update $x_{t+1} = \Pi_{\mathcal{K}}(x_t - \eta g_t)$

8: **end for**

9: Output $\min_{t\in[T]}\{v_t\}$

---

The rest of this section will be dedicated to showing the following main lemma for Algorithm 1.

**Lemma 5.1** (Main, algorithm). *Suppose $\Delta < \frac{\epsilon^2}{16348\sqrt{d}}$, we have: For every $t \in [T]$, if there exists $x^* \in \mathcal{K}$ such that $\tilde{f}(x^*) < \tilde{f}(x_t) - 2\epsilon$, then*

$$\langle -g_t, x^* - x_t \rangle \geq \frac{\epsilon}{64}$$

Assuming this Lemma, we can prove Theorem 3.2.

*Proof of Theorem 3.2.* We first focus on the number of iterations:

For every $t \geq 1$, suppose $\tilde{f}(x^*) < \tilde{f}(x_t) - 2\epsilon$, then we have: (since $\|g_t\| \leq 2d/r \leq \frac{256d}{\epsilon}$)

$$
\begin{aligned}
\|x^* - x_{t+1}\|_2^2 &\leq \|x^* - (x_t - \eta g_t)\|_2^2 \\
&= \|x^* - x_t\|_2^2 - 2\eta\langle x^* - x_t, g_t\rangle + \eta^2\|g_t\|_2^2 \\
&\leq \|x^* - x_t\|_2^2 - \frac{\eta\epsilon}{64} + \eta^2\frac{65536d^2}{\epsilon^2} \\
&\leq \|x^* - x_t\|_2^2 - \frac{\epsilon^4}{8388608d^2} + \frac{\epsilon^4}{4194304d^2} \\
&= \|x^* - x_t\|_2^2 - \frac{\epsilon^4}{8388608d^2}
\end{aligned}
$$

Since originally $\|x^* - x_1\| \leq 1$, the algorithm ends in $poly(d, \frac{1}{\epsilon})$ iterations.

Now we consider the sample complexity. Since we know that

$$
\left\|\frac{d}{r}\tilde{g}(x_t + rw)w\right\|_2 \leq \frac{64d}{\epsilon}
$$

By standard concentration bound we know that we need $poly(d, \frac{1}{\epsilon})$ samples to estimate the expectation up to error $\frac{\epsilon}{2097152}$ per iteration. $\square$

Due to space constraints, we forward the proof of Lemma 5.1 to the appendix.

## 6  Discussion and open problems

### 6.1  Arbitrary Lipschitz constants and diameter

We assumed throughout the paper that the convex function $f$ is 1-Lipschitz and the convex set $\mathcal{K}$ has diameter 1. Our results can be easily extended to arbitrary functions and convex sets through a simple linear transformation. For $f$ with Lipschitz constant $\|f\|_{\mathsf{Lip}}$ and $\mathcal{K}$ with diameter $D$, and the corresponding approximately convex $\tilde{f}$, define $\tilde{g} : \frac{\mathcal{K}}{D} \to \mathbb{R}$ as $\tilde{g}(x) = \frac{1}{D\|f\|_{\mathsf{Lip}}}\tilde{f}(rx)$. (Where $\frac{\mathcal{K}}{D}$ is the rescaling of $\mathcal{K}$ by a factor of $\frac{1}{D}$.) This translates to $\|\tilde{g}(x) - g(x)\|_2 \leq \frac{\Delta}{R\|f\|_{\mathsf{Lip}}}$. But $g(x) = \frac{f(Rx)}{R\|f\|_{\mathsf{Lip}}}$ is 1-Lipschitz over a set $\frac{\mathcal{K}}{R}$ of diameter 1. Therefore, for general functions over a general convex sets, our result trivially implies the rate for being able to optimize approximately-convex functions is

$$
\frac{\Delta}{R\|f\|_{\mathsf{Lip}}} = \max\left\{\frac{1}{\sqrt{d}}\left(\frac{\epsilon}{R\|f\|_{\mathsf{Lip}}}\right)^2, \frac{1}{d}\frac{\epsilon}{R\|f\|_{\mathsf{Lip}}}\right\}
$$

which simplifies to $\Delta = \max\left\{\frac{\epsilon^2}{\sqrt{d}R\|f\|_{\mathsf{Lip}}}, \frac{\epsilon}{d}\right\}$.

### 6.2  Body specific bounds

Our algorithmic result matches the lower bound on well-conditioned bodies. The natural open problem is to resolve the problem for arbitrary bodies. [9]

Also note the lower bound can not hold for any convex body $\mathcal{K}$ in $\mathbb{R}^d$: for example, if $\mathcal{K}$ is just a one dimensional line in $\mathbb{R}^d$, then the threshold should not depend on $d$ at all. But even when the "inherent dimension" of $\mathcal{K}$ is $d$, the result is still body specific: one can show that for $\tilde{f}$ over the simplex in $\mathbb{R}^d$, when $\epsilon \geq \frac{1}{\sqrt{d}}$, it is possible to optimize $\tilde{f}$ in polynomial time even when $\Delta$ is as large as $\epsilon$. [10]

Finally, while our algorithm made use of the well-conditioning – what is the correct property/parameter of the convex body that governs the rate of $T(\epsilon)$ is a tantalizing question to explore in future work.

## Footnotes

[1]The $\tilde{\Omega}$ notation hides $polylog(d/\epsilon)$ factors.

[2]The assumptions on the diameter of $\mathcal{K}$ and the Lipschitz condition are for convenience of stating the results. (See Section **??** to extend to arbitrary diameter and Lipschitz constant)

[3]Though these are not too difficult to derive from the additive ones, considering the convex body has diameter bounded by 1.

[4]Generalizing to arbitrary Lipschitz constants and diameters is discussed in Section 6.

[5]Since we normalize $f$ to be 1-Lipschitz and $\mathcal{K}$ to have diameter 1, the problem is only interesting for $\epsilon \leq 1$

[6]For the proof sketch only, to maintain ease of reading all of the inequalities we state will be only correct up to constants. In the actual proofs we will be completely formal.

[7]This follows from the results in [DKS14]

[8]We pick $\mathbb{B}_{\frac{1}{2}}(0)$ instead of the unit ball in order to ensure the diameter is 1.

[9]We do not show it here, but one can prove the upp/lower bound still holds over the hypercube and when one can find a ball of radius $\epsilon$ that has most of the mass in the convex body $\mathcal{K}$.

[10]Again, we do not show that here, but essentially one can search through the $d + 1$ lines from the center to the $d + 1$ corners.

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
