[Supplementary Material]

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

For convenience of delivering the ideas, we divide the proof into each of these three cases. The proofs are essentially the same in each case, with minor modifications in the calculation.

## 4.2 Case 1: $\frac{1}{\sqrt{d}} \leq \epsilon \leq \frac{1}{(\log d)^2}$

Let us set $\beta = \frac{\sqrt{c \log \frac{d}{\epsilon}}}{\sqrt{d}}$, $\gamma = 10c\epsilon(\log \frac{d}{\epsilon})^{1.5}$, $\alpha = \frac{1}{\log(1/\gamma)}$.

Here, we consider sufficiently large $d_c$ such that when $d \geq d_c$, $\gamma < \frac{1}{2}$ so $\alpha < 1$

Following the the proof outline, we first show the minimum of $f_w$ is small, in particular we will show $f_w(w) \leq -2\epsilon$. Note that $f_w(x) = \max\left\{\tilde{h}_{\mathsf{LCE}}(x), l_w(x)\right\}$ and $l_w(w) = -8\epsilon\|w\|_2^2 = -2\epsilon$, therefore, we can just focus on $\tilde{h}_{\mathsf{LCE}}(w)$:

We will need the following proposition:

**Proposition 4.2.** *Let $\tilde{h}$ be the function defined in Construction 4.1, then we have:*

$$\nabla \tilde{h}(x) = \frac{1+\alpha}{2}\|x\|_2^{\alpha-1}x$$

**Lemma 4.1.** $\tilde{h}_{LCE}(w) \leq \frac{1}{2}(1+\alpha)\beta\gamma^\alpha - \frac{1}{2}\alpha\gamma^{1+\alpha}$

*Proof of Lemma 4.1.* By the definition of LCE, we have:

$$\tilde{h}_{\mathsf{LCE}}(w) = \max_{x \in \mathcal{H}}\{\langle w - x, \nabla\tilde{h}(x)\rangle + \tilde{h}(x)\}$$

$$= \max_{x \in \mathcal{H}}\left\{\left\langle w - x, \frac{1+\alpha}{2}\|x\|_2^{\alpha-1}x\right\rangle + \frac{1}{2}\|x\|_2^{1+\alpha}\right\}$$

$$= \max_{x \in \mathcal{H}}\left\{\frac{1+\alpha}{2}\|x\|_2^{\alpha-1}\langle w, x\rangle - \frac{\alpha}{2}\|x\|_2^{1+\alpha}\right\}$$

$$\leq \max_{x \in \mathcal{H}}\left\{\frac{(1+\alpha)\beta}{2}\|x\|_2^\alpha - \frac{\alpha}{2}\|x\|_2^{1+\alpha}\right\}$$

where the last inequality is due to the fact that $x \in \mathcal{H}$, so: $|\langle x, w\rangle| \leq \beta\|x\|_2$. Now consider the function $g(y) = \frac{(1+\alpha)\beta}{2}y^\alpha - \frac{\alpha}{2}y^{1+\alpha}$: we know that $g'(y) = \frac{(1+\alpha)\alpha\beta}{2}y^{\alpha-1} - \frac{\alpha(1+\alpha)}{2}y^\alpha$. Notice that $g'(y) = \frac{(1+\alpha)}{2}y^{\alpha-1}(\beta - y)$. For $x \in \mathcal{H}$, since $x \notin \mathcal{C}$, we have

$$\|x\|_2 \geq \gamma = 10c\epsilon\left(\log\frac{d}{\epsilon}\right)^{1.5} \geq \frac{10c\left(\log\frac{d}{\epsilon}\right)^{1.5}}{\sqrt{d}} \geq 10\beta \geq \beta$$

Where the second inequality is due to $\epsilon \geq \frac{1}{\sqrt{d}}$ and the second last inequality is due to $c \geq 1, \epsilon \leq 1$.

Therefore, $g'(\|x\|_2) < 0$, which implies that $g(\|x\|_2)$ in $\mathcal{H}$ increases as $\|x\|_2$ decreases. Hence:

$$\tilde{h}_{\mathsf{LCE}}(w) \leq g(\gamma) = \frac{(1+\alpha)\beta\gamma^\alpha}{2} - \frac{\alpha\gamma^{1+\alpha}}{2}$$

$\square$

As a corollary, we get the following:

**Corollary 4.1.** $f_w(w) = -2\epsilon$

*Proof of Corollary 4.1.* Note that $l_w(w) = -2\epsilon$, moreover, since $\gamma^\alpha = \gamma^{\frac{1}{-\log\gamma}} = 2^{\frac{\log\gamma}{-\log\gamma}} = \frac{1}{2}$, have:

$$
\begin{aligned}
\frac{1}{2}(1+\alpha)\beta\gamma^\alpha - \frac{1}{2}\alpha\gamma^{1+\alpha} &= \frac{1}{4}(1+\alpha)\beta - \frac{1}{4}\alpha\gamma \\
&\leq \frac{1}{4}(1+\alpha)\beta - \frac{1}{4}\alpha\gamma
\end{aligned}
$$

By $\gamma \geq 10\beta$ we can conclude:

$$
\frac{1}{2}(1+\alpha)\beta\gamma^\alpha - \frac{1}{2}\alpha\gamma^{1+\alpha} \leq \frac{\beta}{4} - \frac{9}{40}\alpha\gamma
$$

By the definition of $\alpha$, we have: for every $c \geq 1$, there exists $d_c = 1600c$, for every $d \geq d_c$ and every $1 \geq \epsilon \geq \frac{1}{\sqrt{d}}$

$$
\alpha = \frac{1}{\log(1/\gamma)} = \frac{1}{\log\frac{1}{10c\epsilon(\log d/\epsilon)^{1.5}}} \geq \frac{1}{\log d}
$$

Therefore,

$$
\alpha\gamma \geq \frac{10c\epsilon(\log d/\epsilon)^{1.5}}{\log d} \geq 10c\epsilon\sqrt{\log\frac{d}{\epsilon}}
$$

By $\beta = \frac{\sqrt{c\log\frac{d}{\epsilon}}}{\sqrt{d}}$, we can conclude that

$$
\frac{1}{2}(1+\alpha)\beta\gamma^\alpha - \frac{1}{2}\alpha\gamma^{1+\alpha} \leq \frac{\beta}{4} - \frac{9}{40}\alpha\gamma \leq \frac{1}{4}\frac{\sqrt{c\log\frac{d}{\epsilon}}}{\sqrt{d}} - \frac{9}{4}c\epsilon\sqrt{\log\frac{d}{\epsilon}} \leq -2\epsilon
$$

The last inequality is due to $\epsilon \geq \frac{1}{\sqrt{d}}, c \geq 1$.

Which implies $\tilde{h}_{\mathsf{LCE}}(w) \leq -2\epsilon$. This completes the proof. $\qquad\square$

### 4.3  $f_w(x) = \tilde{h}(x)$ in $\mathcal{H}$

We now show that $f_w(x) = \tilde{h}(x)$ in $\mathcal{H}$:

**Lemma 4.2.** *For every $x \in \mathcal{H}$, $f_w(x) = \tilde{h}(x)$*

*Proof of Lemma 4.2.* Note that $\tilde{h}$ is a convex function defined on $\mathcal{H}$, therefore, $\tilde{h}_{\mathsf{LCE}} = \tilde{h}$ on $\mathcal{H}$. Now we consider $l_w$ on $\mathcal{H}$: Since $\forall x \in \mathcal{H}$, $|\langle x, w\rangle| \leq \beta\|x\|_2$, $l_w(x) = -8\epsilon\langle x, w\rangle \leq 8\epsilon\beta\|x\|_2 \leq \frac{1}{2}\gamma^\alpha\|x\|_2$ where the last inequality follows by noticing that for every $c \geq 1$, there exists $d_c = 8192c$, such that for every $d \geq d_c$ and $1 \geq \epsilon \geq \frac{1}{\sqrt{d}}$

$$
8\beta\epsilon \leq 8\beta \leq 8\frac{\sqrt{c\log\frac{d}{\epsilon}}}{\sqrt{d}} \leq 8\frac{\sqrt{c\log(d^{1.5})}}{\sqrt{d}} \leq \frac{1}{4} = \frac{\gamma^\alpha}{2}
$$

Moreover, for $x \in \mathcal{H}$, we know that $\frac{1}{2}\gamma^\alpha\|x\|_2 \leq \tilde{h}(x) = \tilde{h}_{\mathsf{LCE}}(x)$, therefore, $f_w(x) = \tilde{h}(x) = \frac{1}{2}\|x\|_2^{1+\alpha}$. $\qquad\square$

### 4.4  Approximate convexity and constructing $G(x)$

Finally, we show that $\tilde{f}_w$ is indeed a $\Delta$-approximately convex, by showing $\forall x \in \mathcal{K}, |f_w - \tilde{f}_w| \leq \Delta$ and $f_w$ is 1-Lipschitz and convex. Note that by construction, $\tilde{f}_w$ only differs from $f_w$ on $\mathcal{K} \cap \mathcal{C} \cap \overline{\mathcal{A}}$, so it's sufficient to focus on the set $\mathcal{K} \cap \mathcal{C} \cap \overline{\mathcal{A}}$.

We will need the following simple bound on the value of $l_w$ in $\mathcal{K} \cap \mathcal{C} \cap \overline{\mathcal{A}}$.

**Lemma 4.3.** *For every $x \in \mathcal{K} \cap \mathcal{C} \cap \overline{\mathcal{A}}$, we have $|l_w(x)| \leq \frac{1}{2}\Delta$.*

*Proof of Lemma 4.3.* For $x \in \mathcal{K} \cap \mathcal{C} \cap \overline{\mathcal{A}}$, $|l_w(x)| = 8\epsilon|\langle x, w\rangle| \leq 8\epsilon\beta\|x\|_2 \leq 8\epsilon\beta\gamma \leq \frac{1}{2}\Delta$.

The last inequality holds since $\Delta \geq 160\frac{\epsilon^2}{\sqrt{d}}\left(c\log\frac{d}{\epsilon}\right)^2$ and $c \geq 1$ $\qquad\square$

**Proposition 4.3.** *$\tilde{f}_w$ is a $\Delta$-approximately convex.*

*Proof.* First, notice that $f_w$ is convex and 1-Lipschitz.

Since $f_w$ is a point-wise maximum of $\tilde{h}_{\mathsf{LCE}}$ and $l_w$, it is convex.

Furthermore, we claim both $\tilde{h}_{\mathsf{LCE}}$ and $l_w$ are 1-Lipschitz, which will imply that $f_w$ is 1-Lipschitz. Indeed, $l_w$ is 1-Lipschitz by definition, and the norm of the gradient of $\tilde{h}_{\mathsf{LCE}}$ is upper bounded by $\frac{1}{2}(1+\alpha) \leq 1$ since $\alpha < 1$.

Now we argue $\max_{x\in\mathcal{K}}|f_w(x) - \tilde{f}_w(x)| \leq \Delta$.

By Lemma 4.3, we know that when $x \in \mathcal{K} \cap \mathcal{C} \cap \overline{\mathcal{A}}$, $l_w(x) \geq -\frac{1}{2}\Delta$. Since $f_w(x) = \max\{l_w(x), \tilde{h}_{\mathsf{LCE}}(x)\}$, we have $f_w(x) \geq -\frac{1}{2}\Delta$. The claim follows. $\qquad\square$

Now we construct $G(x)$, which does not depend on $w$, we want to show that for an algorithm with small number of queries of $\tilde{f}_w$, it can not distinguish $f_w$ from this function.

**Construction 4.3.** *Let $G : \mathcal{K} \to \mathbb{R}$ be defined as:*

$$G(x) = \begin{cases} \max\left\{\frac{1+\alpha}{4}\|x\|_2 - \frac{\alpha}{4}\gamma, \frac{1}{2}\Delta\right\} & \text{if } x \in \mathcal{K}\cap\mathcal{C}\,; \\ \frac{1}{2}\|x\|_2^{1+\alpha} & \text{otherwise.} \end{cases}$$

**Lemma 4.4.** *$G(x) \geq 0$ and $\{x \in \mathcal{K} \mid G(x) \neq \tilde{f}_w(x)\} \subseteq \mathcal{A}$*

*Proof of Lemma 4.4.* By Lemma 4.2, $\tilde{f}_w(x) = f_w(x) = \tilde{h}(x)$ for $x \in \mathcal{H}$. Moreover, by definition, $\tilde{h}(x) = \frac{1}{2}\|x\|_2^{1+\alpha}$. Therefore, $\tilde{f}_w(x) = G(x)$ for $x \in \mathcal{H}$. So we only need consider $\mathcal{K}\cap\mathcal{C}\cap\overline{\mathcal{A}}$. Note that $|l_w(x)| \leq \frac{1}{2}\Delta$ in $\mathcal{K}\cap\mathcal{C}\cap\overline{\mathcal{A}}$ by Lemma 4.3. Therefore, for $x \in \mathcal{K}\cap\mathcal{C}\cap\overline{\mathcal{A}}$, $\tilde{f}(x) = \max\left\{\tilde{h}_{\mathsf{LCE}}(x), \frac{1}{2}\Delta\right\}$.

We conclude the proof by noticing that for every $x \in \mathcal{K}\cap\mathcal{C}\cap\overline{\mathcal{A}}$ (recall $\mathcal{K}$ the ball of radius $1/2$ centered at 0), there exists $y \in \mathcal{H}$ such that $\|y\|_2 = \gamma$ and $\langle x, y\rangle = \|x\|_2\|y\|_2$. Which implies that

$$\begin{aligned}
\tilde{h}_{\mathsf{LCE}}(x) &= \max_{x'\in\mathcal{H}}\{\langle x - x', \nabla\tilde{h}(x')\rangle + \tilde{h}(x')\} \\
&= \max_{x'\in\mathcal{H}}\left\{\frac{1+\alpha}{2}\|x'\|_2^{\alpha-1}\langle x, x'\rangle - \frac{\alpha}{2}\|x'\|_2^{1+\alpha}\right\} \\
&= \frac{1+\alpha}{2}\|y\|_2^{\alpha-1}\langle x, y\rangle - \frac{\alpha}{2}\|y\|_2^{1+\alpha} \\
&= \frac{1+\alpha}{4}\|x\|_2 - \frac{\alpha}{4}\gamma
\end{aligned}$$

Where the third equality follows from the following observations:

(1). When $\|x'\|_2$ is fixed, the best $x'$ should be aligned with $x$: $\langle x', x\rangle = \|x'\|_2\|x\|_2$.

(2). defining

$$g(s) = \frac{1+\alpha}{2}\|x\|_2 s^\alpha - \frac{\alpha}{2}s^{1+\alpha}$$

We have:

$$g'(s) = \frac{(1+\alpha)\alpha(\|x\|_2 - s)}{2}s^{\alpha-1} < 0$$

For $\|x\|_2 \leq \gamma \leq s$. $\qquad\square$

## 4.5   Putting everything together

*Proof of Theorem 3.1 for $\frac{1}{\sqrt{d}} \le \epsilon \le \frac{1}{(\log d)^2}$.* With everything prior to this set up, the final claim is somewhat standard. We want to show that no algorithm can, with probability $\ge \frac{1}{2}$, output a point $x$, s.t. $\tilde{f}_w(x) \le \min_x \tilde{f}_w(x) + \epsilon$. Since we know that $\tilde{f}_w(x)$ agrees with $G(x)$ everywhere except in $\mathcal{K} \cap \mathcal{A}$, and $G(x)$ satisfies $\min_x G(x) \ge \min_x \tilde{f}_w(x) + \epsilon$, we only need to show that with high probability, any polynomial time algorithm will not query any point in $\mathcal{K} \cap \mathcal{A}$.

Consider a (potentially) randomized algorithm $A$, making random choices $R_1, R_2, \ldots, R_m$. Conditioned on a particular choice of randomness $r_1, r_2, \ldots, r_m$, for a random choice of $w$, each $r_i$ lies in $\mathcal{A}$ with probability at most $\exp(-c \log(d/\epsilon))$, by a standard Gaussian tail bound. Union bounding, since $m = o((\frac{d}{\epsilon})^c)$ for an algorithm that runs in time $o((\frac{d}{\epsilon})^c)$, the probability that at least of the queries of $A$ lies in $\mathcal{A}$ is at most $\frac{1}{2}$.

But the claim is true for any choice $r_1, r_2, \ldots, r_m$ of the randomness, by averaging, the claim holds for $r_1, r_2, \ldots, r_m$ being sampled according to the randomness of the algorithm.

$\square$

## 4.6   Case 2: $\epsilon \le \frac{1}{\sqrt{d}}$

In the case where $\epsilon \le \frac{1}{\sqrt{d}}$, the proof proceeds exactly the same as before, but with a different setting of the parameters. In this case we set $\beta = \frac{\sqrt{c \log d/\epsilon}}{\sqrt{d}}$, $\gamma = \frac{10c}{\sqrt{d}}(\log d/\epsilon)^{3/2}$, and $\alpha = \frac{1}{\log(1/\gamma)}$.

We proceed to verify that each Lemma still holds under this parameter setting. We first show that Lemma 4.1 still holds.

*Proof of Lemma 4.1.* Following the same calculation as in previous section, it is enough to show that $\gamma \ge \beta$ in this setting. We can check that for every $c \ge 1$

$$\frac{\beta}{\gamma} = \frac{\sqrt{c \log d/\epsilon}}{10c(\log d/\epsilon)^{1.5}} = \frac{\sqrt{c}}{10c \log d/\epsilon} < \frac{\sqrt{c}}{10c} \le \frac{1}{10} < 1$$

$\square$

We then check that Corollary 4.1 still holds.

*Proof of Corollary 4.1.* Following the same calculation in the previous section, it is sufficient to show that

$$\frac{\beta}{4} - \frac{9}{40}\alpha\gamma \le -2\varepsilon$$

Putting in the specific numbers, we obtain: since $\epsilon \le \frac{1}{\sqrt{d}}$

$$\beta = \frac{\sqrt{c \log d/\epsilon}}{\sqrt{d}}$$

On the other hand, we have:

$$\alpha\gamma \ge \frac{10c(\log d/\epsilon)^{1.5}}{\sqrt{d}\log\sqrt{d/\epsilon}} \ge \frac{10c\sqrt{\log d/\epsilon}}{\sqrt{d}}$$

Therefore, for $c \ge 1$

$$\frac{\beta}{4} - \frac{9}{40}\alpha\gamma \le \frac{\sqrt{c \log d/\epsilon}}{4\sqrt{d}} - \frac{9c\sqrt{\log d/\epsilon}}{4\sqrt{d}} \le -\frac{2}{\sqrt{d}} \le -2\epsilon$$

$\square$

We then check that Lemma 4.2 still holds.

*Proof of Lemma 4.2.* Following the same calculation in the previous section, it is sufficient to show that

$$8\beta\epsilon \leq \frac{1}{4}$$

Putting in the specific bound, we know that for every $c \geq 0$, there exists a $d_c = 32c^2$ such that for every $d \geq d_c$,

$$8\beta\epsilon = 8\frac{\sqrt{c\log d/\epsilon}}{\sqrt{d}}\epsilon \leq \frac{1}{4}$$

$\square$

It remains to check that Lemma 4.3 holds.

*Proof of Lemma 4.3.* Following the same calculation in the previous section, it is sufficient to show that

$$8\epsilon\beta\gamma \leq \frac{1}{2}\Delta$$

Putting in the specific bound, we know that

$$8\beta\epsilon\gamma = \frac{80c^{1.5}\epsilon(\log d/\epsilon)^2}{d} \leq \frac{\Delta}{2}$$

Where the last inequality follows from $\Delta \geq \frac{\epsilon}{d}(13c\log d/\epsilon)^2$. $\square$

Note that Lemma 4.4 holds regardless of the choice of $\alpha, \beta, \gamma$.

### 4.7 Case 3: $\frac{1}{64} \geq \epsilon \geq \frac{1}{(\log d)^2}$

In this case, we can choose $\beta = \frac{\sqrt{c\log d}}{\sqrt{d}}$, $\gamma = \frac{1}{2}$ and $\alpha = 1$. Actually, since $\mathcal{C} = \mathcal{K}$ in this case, it reduces to having only linear function $l_w$.

We can check that for sufficiently large $d_c = 4096c^2$, for every $d \geq d_c$, we have: $\beta \leq \frac{\gamma}{10}$; $\frac{\beta}{4} - \frac{9}{40}\alpha\gamma \leq -\frac{1}{10} \leq -2\epsilon$, $8\beta\epsilon \leq \frac{8\sqrt{c\log d}}{\sqrt{d}} \leq \frac{1}{4}$ and $8\beta\epsilon\gamma = \frac{4\epsilon\sqrt{c\log d}}{\sqrt{d}} \leq \frac{1}{2}\Delta$. So all the Lemma 4.1, 4.1, 4.2, 4.3 holds. Which completes the proof.

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

*Proof of Lemma 5.1.* Proceeding towards applying the FKM framework, we have, since $|g(x) - h(x)| \leq \Delta$:

$$\mathbb{E}_{w \sim U(S^d)} \left[ \left\langle -\frac{d}{r} g(x_t + rw)w, e \right\rangle \right] \geq \mathbb{E}_{w \sim U(S^d)} \left[ \left\langle -\frac{d}{r} h(x_t + rw)w, e \right\rangle \right] - \frac{d}{r} \Delta \mathbb{E}_{w \sim U(S^d)} |\langle w, e \rangle|$$

We want to lower bound the quantity on the RHS by $\frac{\epsilon}{64}$. Since the absolute value of the second term is upper bounded by $\frac{d}{r} \Delta (2d^{-1/2}) \leq \frac{\epsilon}{32}$, it is enough to bound the first term by $\frac{\epsilon}{16}$.

Let's proceed to the first term. By applying Stokes' theorem, we have:

$$\mathbb{E}_{w \sim U(S^d)} \left[ \left\langle -\frac{d}{r} h(x_t + rw)w, e^* \right\rangle \right]$$

$$= -\left\langle \frac{1}{\mathsf{Vol}[\mathbb{B}_1(0)]} \int_{z \in \mathbb{B}_1(0)} \nabla h(x_t + rz)dz, e^* \right\rangle$$

$$= -\frac{1}{\mathsf{Vol}[\mathbb{B}_1(0)]} \int_{z \in \mathbb{B}_1(0)} \langle \nabla h(x_t + rz), e^* \rangle \, dz$$

In order to evaluate gradients of $h$, we will need to develop machinery for dealing with the projections.

Note the following observation: for any point $y \in \mathbb{R}^d$, $x = \Pi_\mathcal{K}(y)$ is given by the solution to the system of equations in $x, \lambda$:

$$x + \lambda \nabla F(x) = y, \quad \lambda = \frac{\|y - x\|_2}{\|\nabla F(x)\|_2} \tag{1}$$

For simplicity, we denote $x_t$ to $x$ and let $e = x^* - x$.

Denoting $y_s = y + se, x_s = \Pi_\mathcal{K}(y_s)$, we have using (1) and the chain rule:

$$e = \frac{\partial y_s}{\partial s} = \frac{\partial x_s}{\partial s} + \frac{\partial \lambda_s}{\partial s} \nabla F(x) + \lambda_s \nabla^2 F(x) \frac{\partial x_s}{\partial s} \tag{2}$$

We will be evaluating all the partial derivatives at $s = 0$, so as a shorthand, let us denote by $\frac{\partial \lambda}{\partial s}, \frac{\partial x}{\partial s}$ the quantities $\frac{\partial \lambda_s}{\partial s}\Big|_{s=0}, \frac{\partial x_s}{\partial s}\Big|_{s=0}$.

Towards calculating $\frac{\partial h(y_s)}{\partial s}$, we proceed to calculate $\frac{\partial f(x_s)}{\partial s}$ and $\frac{\partial \|y_s - x_s\|_2}{\partial s}$.

For $\frac{\partial f(x_s)}{\partial s}$, by (2) we have:

$$\frac{\partial f(x_s)}{\partial s}\Big|_{s=0} = \left\langle \nabla f(x), \frac{\partial x_s}{\partial s} \right\rangle = \langle \nabla f(x), e \rangle - \frac{\partial \lambda}{\partial s} \langle \nabla f(x), \nabla F(x) \rangle - \lambda \nabla f(x)^\top \nabla^2 F(x) \frac{\partial x}{\partial s} \tag{3}$$

On the other hand, we wish to show $\frac{\|y_s - x_s\|_2}{\partial s}\Big|_{s=0} = \frac{1}{\|\nabla F(x)\|_2} \langle \nabla F(x), e \rangle$. Using the fact that for any differentiable function $g(x)$, $\frac{d}{dx} \|g(x)\|_2 = \|g(x)\|_2^{-1} \frac{d}{dx} g(x)$ we have

$$\frac{\|y_s - x_s\|_2}{\partial s}\Big|_{s=0} = \|y - x\|_2^{-1} \left\langle y - x, \frac{\partial y}{\partial s} - \frac{\partial x}{\partial s} \right\rangle$$

$$= \frac{\partial \lambda}{\partial s} \|\nabla F(x)\|_2 + \frac{\lambda}{\|\nabla F(x)\|_2} \nabla F(x)^\top \nabla^2 F(x) \frac{\partial x}{\partial s} \tag{4}$$

where the second equality follows since by (1), we have $\frac{y - x}{\|y - x\|_2} = \frac{\nabla F(x)}{\|\nabla F(x)\|_2}$.

Since $F(x_s) = 0$ by taking gradients on both sides we have $\langle \nabla F(x), \frac{\partial x}{\partial s} \rangle = 0$ which gives us by (1):

$$\langle \nabla F(x), e \rangle = \frac{\partial \lambda}{\partial s} \|\nabla F(x)\|_2^2 + \lambda \nabla F(x)^\top \nabla^2 F(x) \frac{\partial x}{\partial s} \tag{5}$$

Combine with Equality (4) we get: $\left. \frac{\|y_s - x_s\|_2}{\partial s} \right|_{s=0} = \frac{1}{\|\nabla F(x)\|_2} \langle \nabla F(x), e \rangle$ as we wanted.

For notational convenience, let's denote $\nabla \tilde{F}(x) = \frac{\nabla F(x)}{\|\nabla F(x)\|_2}$. From the above estimates, putting (3) and (4) together we have:

$$\begin{aligned} \frac{h(y_s)}{\partial s} &= \langle \nabla f(x), e \rangle + \langle \nabla \tilde{F}(x), e \rangle (1 - \langle \nabla f(x), \nabla \tilde{F}(x) \rangle) \\ &\quad + \lambda \left( \nabla \tilde{F}(x) \langle \nabla \tilde{F}(x), f(x) \rangle - \nabla f(x) \right)^\top \nabla^2 F(x) \frac{\partial x}{\partial s} \end{aligned} \tag{6}$$

We will bound each of the terms on the RHS individually. Namely, we show:

$$\langle \nabla f(x), e \rangle \leq -\frac{\epsilon}{2} \tag{7}$$

$$\langle \nabla \tilde{F}(x), e \rangle (1 - \langle \nabla f(x), \nabla \tilde{F}(x) \rangle) \leq 0 \tag{8}$$

$$\lambda \left( \nabla \tilde{F}(x) \langle \nabla \tilde{F}(x), f(x) \rangle - \nabla f(x) \right)^\top \nabla^2 F(x) \frac{\partial x}{\partial s} \leq -\frac{\epsilon}{8} \tag{9}$$

Proceeding to (7), since $f(x^*) \leq f(x) - \epsilon$, $\|x - x^*\|_2 \leq r$ and $f$ is 1-Lipschitz, we know that $f(x^*) \leq f(x) - \frac{\epsilon}{2}$. By convexity of $f$, we have

$$f(x^*) \geq f(x) + \langle \nabla f(x), x^* - x \rangle$$

which by simple rearranging gives $\langle \nabla f(x), e \rangle \leq -\frac{\epsilon}{2}$.

For (8), we know that $x^*$ lies in $\mathcal{K}$, hence using the fact that $\nabla F(x)$ is a normal vector to $\mathcal{K}$ at $x$, we get $\langle \nabla F(x), e \rangle \leq 0$. Furthermore, since $|\langle \nabla f(x), \nabla \tilde{F}(x) \rangle| \leq 1$, we know that $(1 - \langle \nabla f(x), \nabla F(x) \rangle) \in [0, 2]$, which implies that

$$\langle \nabla \tilde{F}(x), e \rangle \left( 1 - \langle \nabla f(x), \nabla \tilde{F}(x) \rangle \right) \leq 0$$

Finally, consider the last term. By (2) and (5) we get:

$$\begin{aligned} (I + \lambda \nabla^2 F(x)) \frac{\partial x_s}{\partial s} &= e - \frac{\partial \lambda}{\partial s} \nabla F(x) \\ &= e - \langle \nabla \tilde{F}(x), e \rangle \nabla \tilde{F}(x) + \langle \nabla \tilde{F}(x), e \rangle \nabla \tilde{F}(x) - \frac{\partial \lambda}{\partial s} \nabla F(x) \\ &= (e - \langle \nabla \tilde{F}(x), e \rangle \nabla \tilde{F}(x)) + \lambda \nabla \tilde{F}(x)^\top \nabla^2 F(x) \frac{\partial x_s}{\partial s} \nabla \tilde{F}(x) \end{aligned}$$

Multiplying on the left and right by $(I - \nabla \tilde{F}(x) \nabla \tilde{F}(x)^\top)$ on both sides of the above equality, and using the fact that $\nabla \tilde{F}(x)^\top \frac{\partial x_s}{\partial s} = 0$ and $(I - \nabla \tilde{F}(x) \nabla \tilde{F}(x)^\top) \nabla \tilde{F}(x) = 0$, we have:

$$(I + \lambda (I - \nabla \tilde{F}(x) \nabla \tilde{F}(x)^\top) \nabla^2 F(x)) \frac{\partial x_s}{\partial s} = (I - \nabla \tilde{F}(x) \nabla \tilde{F}(x)^\top)(e - \langle \nabla \tilde{F}(x), e \rangle \nabla \tilde{F}(x))$$

Denoting $A = (I - \nabla \tilde{F}(x) \nabla \tilde{F}(x)^\top)$, $B = A \nabla^2 F(x)$, we get:

$$A \nabla^2 F(x) \frac{\partial x}{\partial s} = B(I + \lambda B)^{-1} A e$$

We proceed to bound the spectral norm of the RHS (which of course will imply a spectral norm bound on the LHS). Towards that, we first show $\|\lambda B\|_2 \leq \frac{1}{2}$: indeed, by our choice of $r$, we have $\lambda \|\nabla F(x)\|_2 \leq r \leq \frac{1}{2\mu}$. This implies

$$\|\lambda B\|_2 \leq \lambda \|A\|_2 \|\nabla^2 F(x)\|_2 \leq \|\lambda\|_2 \|\nabla F(x)\|_2 \frac{\|\nabla^2 F(x)\|_2}{\|\nabla F(x)\|_2} \leq r\mu \leq \frac{1}{2} \tag{10}$$

where the first inequality follows by submultiplicativity of the spectral norm, and the second by $\|A\| \leq 1$. Therefore we have:

$$\|B(I + \lambda B)^{-1} A e\|_2 \leq 2\|B\|_2 \leq 2\|A\|_2 \|\nabla^2 F(x)\|_2 \leq 2\mu \|\nabla F(x)\|_2$$

where the first inequality and second inequality follow by (10) and the submultiplicativity of the spectral norm; the third is by well-conditioning of the convex body. Finally, this implies $|\nabla f(x)^\top (I - \nabla \tilde{F}(x)\nabla \tilde{F}(x)^\top)\nabla^2 F(x)\frac{\partial x}{\partial s}| \leq 2\mu \|\nabla F(x)\|_2$

Putting (7), (8), (9) together, we get $\left.\frac{h(y_s)}{\partial s}\right|_{s=0} \leq -\frac{\epsilon}{8}$.

Using the above fact and the 1-Lipschitzness of $h$, we get

$$\langle \nabla h(y), e^* \rangle = \langle \nabla h(y), e \rangle + \langle \nabla h(y), e^* - e \rangle \leq -\frac{\epsilon}{8} + r \leq -\frac{\epsilon}{16}$$

Which completes the proof.

$\square$

# 6 Discussion and open problems

## 6.1 Arbitrary Lipschitz constants and diameter

We assumed throughout the paper that the convex function $f$ is 1-Lipschitz and the convex set $\mathcal{K}$ has diameter 1. Our results can be easily extended to arbitrary functions and convex sets through a simple linear transformation. For $f$ with Lipschitz constant $\|f\|_{\mathsf{Lip}}$ and $\mathcal{K}$ with diameter $D$, and the corresponding approximately convex $\tilde{f}$, define $\tilde{g} : \frac{\mathcal{K}}{D} \to \mathbb{R}$ as $\tilde{g}(x) = \frac{1}{D\|f\|_{\mathsf{Lip}}}\tilde{f}(rx)$. (Where $\frac{\mathcal{K}}{D}$ is the rescaling of $\mathcal{K}$ by a factor of $\frac{1}{D}$.) This translates to $\|\tilde{g}(x) - g(x)\|_2 \leq \frac{\Delta}{R\|f\|_{\mathsf{Lip}}}$. But $g(x) = \frac{f(Rx)}{R\|f\|_{\mathsf{Lip}}}$ is 1-Lipschitz over a set $\frac{\mathcal{K}}{R}$ of diameter 1. Therefore, for general functions over a general convex sets, our result trivially implies the rate for being able to optimize approximately-convex functions is

$$\frac{\Delta}{R\|f\|_{\mathsf{Lip}}} = \max\left\{ \frac{1}{\sqrt{d}}\left(\frac{\epsilon}{R\|f\|_{\mathsf{Lip}}}\right)^2, \frac{1}{d}\frac{\epsilon}{R\|f\|_{\mathsf{Lip}}} \right\}$$

which simplifies to $\Delta = \max\left\{ \frac{\epsilon^2}{\sqrt{d}R\|f\|_{\mathsf{Lip}}}, \frac{\epsilon}{d} \right\}$.

## 6.2 Body specific bounds

Our algorithmic result matches the lower bound on well-conditioned bodies. The natural open problem is to resolve the problem for arbitrary bodies. [9]

Also note the lower bound can not hold for any convex body $\mathcal{K}$ in $\mathbb{R}^d$: for example, if $\mathcal{K}$ is just a one dimensional line in $\mathbb{R}^d$, then the threshold should not depend on $d$ at all. But even when the "inherent dimension" of $\mathcal{K}$ is $d$, the result is still body specific: one can show that for $\tilde{f}$ over the simplex in $\mathbb{R}^d$, when $\epsilon \geq \frac{1}{\sqrt{d}}$, it is possible to optimize $\tilde{f}$ in polynomial time even when $\Delta$ is as large as $\epsilon$. [10]

Finally, while our algorithm made use of the well-conditioning – what is the correct property/parameter of the convex body that governs the rate of $T(\epsilon)$ is a tantalizing question to explore in future work.

## Footnotes

[1]The $\tilde{\Omega}$ notation hides $polylog(d/\epsilon)$ factors.

[2]The assumptions on the diameter of $\mathcal{K}$ and the Lipschitz condition are for convenience of stating the results. (See Section **??** to extend to arbitrary diameter and Lipschitz constant)

[3]Though these are not too difficult to derive from the additive ones, considering the convex body has diameter bounded by 1.

[4]Generalizing to arbitrary Lipschitz constants and diameters is discussed in Section 6.

[5]Since we normalize $f$ to be 1-Lipschitz and $\mathcal{K}$ to have diameter 1, the problem is only interesting for $\epsilon \leq 1$

[6] For the proof sketch only, to maintain ease of reading all of the inequalities we state will be only correct up to constants. In the actual proofs we will be completely formal.

[7]This follows from the results in [DKS14]

[8]We pick $\mathbb{B}_{\frac{1}{2}}(0)$ instead of the unit ball in order to ensure the diameter is 1.

[9]We do not show it here, but one can prove the upp/lower bound still holds over the hypercube and when one can find a ball of radius $\epsilon$ that has most of the mass in the convex body $\mathcal{K}$.

[10]Again, we do not show that here, but essentially one can search through the $d + 1$ lines from the center to the $d + 1$ corners.