[Reviews · NeurIPS 2016]

Reviewer 1

Summary

This paper studies the problem of minimizing approximately convex functions by zero order methods. This problem adds one extra layer of difficulty to the traditional exact minimization methods, given by the (uniform) distance ∆ of the objective to a convex 1-Lipschitz function. The main contribution is providing tight upper and lower bounds on the complexity of this problem for all regimes of inexactness ∆, accuracy ε, and dimension d. To the best of my knowledge, all lower bounds (except for Case 3, which is somewhat simpler) are new, and the same is true for the algorithm proposed for the regime ε/d ≼ ∆ ≼ ε^2/\sqrt{d}. This paper considerably extends the understanding of approximate convex optimization by some novel ideas: • The lower bound consists on hiding a linear function on a random slab. The improvement achieved in the paper relies on constructing a convex function outside the slab, which is key on the new results (previous extensions were only linear). • The upper bound is based on stochastic smoothing and gradient methods. By the divergence theorem, one can obtain an approximate gradient by estimating finite differences on random unit vectors, and this approximation is robust to the inexactness introduced by ∆. Although this idea is not entirely new (e.g., the cited work of Shamir), the analysis itself seems to be original, and it is studied in the inexact setup, which is certainly new. Overall, I believe this paper is very interesting, and therefore I strongly recommend its publication at NIPS. However, I have to mention some very mild criticism, particularly on the related literature and the writing style.

Qualitative Assessment

First I would like to justify my ratings for this paper. I believe the quality of the paper is indeed high: it resolves open questions from the literature, it brings new ideas in the construction of hard instances for convex programming, and can hopefully impact related areas. However, there are some aspects that deserve more attention, particularly I think writing requires some extra effort. I would like to describe some high-level criticism for the work: 1. The paper emphasizes the differences between approximately convex minimization and various oracle models; however, it is not mentioned that approximately convex minimization is equivalent to minimization of convex functions with an inexact zero order method. This is somehow a more general setup (as the oracle does not need to be consistent with information among iterations) and it is arguably simpler to grasp. 3. The paper needs some writing revision. Aside from correcting typos, it would be nice to give some insights on the intriguing new regime ∆ = O( ε^2 / sqrt{d}), and perhaps expand the motivating discussions on proof techniques (Section 3.1) and the informal discussion on the lower bound, at the beginning of Section 4. Below are some minor comments: 1. The Abstract and first lines of the Introduction stress the relevance of optimization with nonconvex objectives, mentioning applications such as neural networks, graphical models, clustering, etc. Algorithms for minimizing approximately convex functions are not applicable to these problems, so it seems inappropriate to mention these applications. 2. In the paper it should be mentioned the very first reference on inexact zero order methods from the Nemirovsky and Yudin monograph (Exercise 1 in page 360). This example provides the same superpolynomial lower bound when ∆ = Ω(ε/ sqrt{d}) (in line 77) for the unit Euclidean ball (instead of the hypercube), and it fits precisely within the model of the paper. 3. Line 145. The definition of the lower convex envelope is somewhat confusing. First, some form of differentiability is required so that the formula makes sense. Second, the remark: “When the input function F is convex, LCE(F) can be considered an extension of F to the entire R^d” is used incorrectly, as it is later used for a function defined on a nonconvex set (first line of proof of Lemma 4.2). I suggest to prove “by hand” that LCE and the function coincide in the proof, as it is easy and does not lead to ambiguities. 4. Line 282. “Thus, for every point \tilde u_1, there exists[..]”. Please add more details on this, as it is an important technical step in the proof.

Confidence in this Review

3-Expert (read the paper in detail, know the area, quite certain of my opinion)


Reviewer 2

Summary

The paper shows how to find a point that achieves close to a minimum for an approximately convex function. If the function is pointwise(additively) Delta close to a 1-Lipschitz convex function, then they give an algorithm that is within epsilon of the minimum value, as long as Delta = O(max{epsilon/d, epsilon^2/sqrt(d)) where d is the dimensionality of the space. The algorithm is only allowed oracle point queries to f. They also show that the trade-off between Delta and epsilon is tight upto logarithmic factors.

Qualitative Assessment

The problem is an important one -- but it is not new and has been studied in a few papers before. As stated in the paper, one part of the upper bound -- when Delta =O(epsilon/d) -- was already known and was based on simulated annealing. The new part here is for the case when Delta closer to epsilon^2/\sqrt{d} which is based on entirely different (perhaps simpler?) techniques for estimating the gradient from [FKM05] that provided an online algorithm for convex optimization in bandit setting. The paper is well written and all the high level details are explained in reasonable clarity before delving into the formal proofs. Another interesting question in this line would be: what if you are given oracle access to gradient of f? Would the number of queries required be lower? Line 187: Change epsilon^2/d to epsilon^2/\sqrt{d} Line 193: There is an extra parenthesis in the last line on page 6.

Confidence in this Review

2-Confident (read it all; understood it all reasonably well)


Reviewer 3

Summary

This paper studies the problem of optimizing a function that is approximately convex in that it respects the convexity condition up to some small additive \delta. It was recently shown by Singer and Vondrak in NIPS 2015 that for \delta in \sqrt d no poly-time algorithm exists for minimizing the function up to a constant error, while Belloni et al. show that when \delta in O(\epsilon/d) an additive \epsilon approximation is possible. This paper closes the gap between these two results.

Qualitative Assessment

For a given accuracy epsilon, the goal is to obtain a solution whose value is within an additive epsilon of the optimal value in time polynomial in the dimension d and 1/epsilon. The paper considers zero-order optimization, in which the function is only accessed through value queries (for example, it is not assumed that the gradient can be computed; it might not even exist since the approximately convex function could even be discontinuous) It is intuitively clear that as delta grows larger compared to epsilon, the problem becomes harder. More precisely, the goal is to find a threshold function T, such that when delta = O(T(epsilon)), then the problem is solvable in polynomial time and when delta = Omega(T(epsilon)), the problem is not solvable in polynomial time (the problem is always solvable in exponential time by considering a square grid of width 1/epsilon). The authors show that this function is T(epsilon) = max(epsilon^2/sqrt{d}, epsilon/d). More precisely: * they provide an information theoretic lower bound, showing that when delta = Omega(T(epsilon)) (up to logarithmic factor), no algorithm making polynomially evaluations of the function can optimize it to precision epsilon. The lower bound relies on the careful construction of a family of functions defined on the unit ball which behaves like ||x||^{1+alpha} unless x lies in a small angle around a random chosen direction. In this small angle, the function can take significantly smaller values, but with very high probability, an algorithm never evaluates the function in this small angle. * they give an algorithm which provably finds an epsilon-approximate solution in the regime where delta = Omega(epsilon/d) and delta = O(epsilon^2/d). Together with a previous algorithm from Belloni et al. in the regime delta = O(epsilon/d), this completes the algorithmic upper bound. Their algorithm uses a natural idea from Flaxman et al., where the gradient of the underlying convex function at some point x is estimated by sampling points in a ball around x. The algorithm is then a gradient descent using this estimated gradient. The analysis relies on showing that even with a delta-approximately convex function, this way of estimating the gradient still provides a sufficiently good descent direction. The construction of the information theoretic lower bound is novel and non-trivial. While the algorithm is inspired by Flaxman et al., its analysis for approximately convex functions is novel.

Confidence in this Review

3-Expert (read the paper in detail, know the area, quite certain of my opinion)


Reviewer 4

Summary

For any real n-dimensional convex function there exists an algorithm that finds a 1-lipshitz function approximation, named the information upper bound, in polynomial time. However, there exist no alogorithm which can find an approximation for the lower bound within polynomial time.

Qualitative Assessment

The overall paper makes sense as to establishing a higher and a lower information limit to find the point of tight convergence window. However, I do not think I can accurately verify all the math proofs because this is not primary research domain. This makes it difficult for me to properly recommend the paper for publication.

Confidence in this Review

1-Less confident (might not have understood significant parts)


Reviewer 5

Summary

This paper provides tight conditions for which approximately-convex functions could be optimized to approximate accuracy. In particular, it gave a optimal rate of approximately convex level (Delta) as a function of final error level (epsilon). This rate is optimal in the sense that for Delta at that range, any algorithms would use super polynomial number of evaluations of f, conversely, an algorithm is proposed to achieve the upper bound with polynomial evaluations. The main theoretical contribution is the construction of the information theoretical lower bound and the design of algorithm that achieves the upper bound.

Qualitative Assessment

The authors explain the main proof techniques and how it is related to previous work before diving into the detailed proof, making the paper easy to understand. The presentation is very clear, except for one minor typo: - Line 185: "ommited" to "omitted" - The theoretical question is very interesting and the authors gave a tight rate. The theoretical question is motivated by the training of neural networks and graphical models, saying that they are non-convex objectives. However, it is not clear how do these popular machine learning problems fall into the non-convex category that this paper considers. It would be interesting for the authors to elaborate a little bit on this point.

Confidence in this Review

2-Confident (read it all; understood it all reasonably well)


Reviewer 6

Summary

This paper studies the lower bounds for approximately convex optimization. The emphasis is on the existence of a polynomial time algorithm in key parameters. Main results are the lower bounds that match the existing upper bounds in the considered setting.

Qualitative Assessment

I apologize that I'm not an expert of the lower bounds on optimization problems, and as a result I cannot give too many constructive comments. First, regarding the technical soundness. I've tried my best to trace the proofs. The presentation, however, rendered the proofs rather difficult to read. The construction in Section 4 seems to be correct as I did not discover any critical mistake. However, I can merely check calculations as the authors did not explain ideas clearly (for example, what role does the LCE play in the construction?). Section 5 is not readable for me. The section starts with a key formula without explaining what the 'r' there is. Later on it is referred to as a tunable parameter, but then the key step (line 254) cites a previous paper, in which I could not directly derive the claimed result. Please at least cite the part where the authors used in the calculation of line 254. Without better writing, I will have to stay conservative on the correctness of the results in Section 5. Secondly, I am not very convinced by the "motivation" in this paper. The authors cited problems like training neural networks, training graphical models, and solving maximum likelihood estimators (of what?). As far as I know, none of these are 'approximately convex' in any sense. For example, loss functions in neural networks present perhaps one of the most non-convex problems nowadays, and I doubt that the results in this paper have any insight into optimizing such functions. In previous works on approximated convex programming, the main motivation is two-stage stochastic programming and zeroth order stochastic convex optimization. It would be good if the authors can look into those applications and draw some conclusions. That would make the importance of this paper much more convincing. Lastly, the presentation of this paper has a huge room for improvement. Typos are everywhere; I will list a few below, and I believe there're many others. Organization of Section 4 is random, proof ideas are not stated clearly, and inequalities are correct only "up to a constant". Similar comments apply to Section 5, although I did not read it in full detail. Things only get worse when it comes to the full length version. In short, this paper seems to contain interesting results. However, the novelty of the proof idea requires other experts to verify, and I feel that this paper needs a major rewriting to meet the NIPS' standard. Here are some typos I found. Please look at the full length version. Line 11: Please use dot or semicolon before the 'however' Line 24: A missing comma before etc. Line 25: The 'are' should be 'is'. Line 96, 126, 189: 'It's' Line 105: The 'every' should be 'ever'. Line 134: In the definition of \mathcal{A}, please write x \in \mathbb{B}_1(0) Line 174: A missing dot. Line 175: The comma after -2\epsilon should be a dot. Line 176: Please add a 'we' in the beginning. Line 179: Take a look at the beginning. Line 187: An extra 'a' before \Delta. Line 222: A random choice over what measure? Line 223: With probability with respect to what measure? The last sentence in the second paragraph of Section 4.5 is weird. Line 250: 'robustNESS'.

Confidence in this Review

2-Confident (read it all; understood it all reasonably well)